

# Human-robot interaction: the impact of robotic aesthetics on anticipated human trust

Joel Pinney, Fiona Carroll and Paul Newbury

Cardiff School of Technologies, Cardiff Metropolitan University, Cardiff, Wales

Corresponding author
Joel Pinney,
st20102131@outlook.cardiffmet.ac.uk

## ABSTRACT

**Background**. Human senses have evolved to recognise sensory cues. Beyond our perception, they play an integral role in our emotional processing, learning, and interpretation. They are what help us to sculpt our everyday experiences and can be triggered by aesthetics to form the foundations of our interactions with each other and our surroundings. In terms of Human-Robot Interaction (HRI), robots have the possibility to interact with both people and environments given their senses. They can offer the attributes of human characteristics, which in turn can make the interchange with technology a more appealing and admissible experience. However, for many reasons, people still do not seem to trust and accept robots. Trust is expressed as a person's ability to accept the potential risks associated with participating alongside an entity such as a robot. Whilst trust is an important factor in building relationships with robots, the presence of uncertainties can add an additional dimension to the decision to trust a robot. In order to begin to understand how to build trust with robots and reverse the negative ideology, this paper examines the influences of aesthetic design techniques on the human ability to trust robots.

**Method**. This paper explores the potential that robots have unique opportunities to improve their facilities for empathy, emotion, and social awareness beyond their more cognitive functionalities. Through conducting an online questionnaire distributed globally, we explored participants ability and acceptance in trusting the Canbot U03 robot. Participants were presented with a range of visual questions which manipulated the robot's facial screen and asked whether or not they would trust the robot. A selection of questions aimed at putting participants in situations where they were required to establish whether or not to trust a robot's responses based solely on the visual appearance. We accomplished this by manipulating different design elements of the robots facial and chest screens, which influenced the human-robot interaction.

**Results**. We found that certain facial aesthetics seem to be more trustworthy than others, such as a cartoon face versus a human face, and that certain visual variables (*i.e.,* blur) afforded uncertainty more than others. Consequentially, this paper reports that participant's uncertainties of the visualisations greatly influenced their willingness to accept and trust the robot. The results of introducing certain anthropomorphic characteristics emphasised the participants embrace of the uncanny valley theory, where pushing the degree of human likeness introduced a thin line between participants accepting robots and not. By understanding what manipulation of design elements created the aesthetic effect that triggered the affective processes, this paper further enriches our knowledge of how we might design for certain emotions, feelings, and ultimately more socially acceptable and trusting robotic experiences.

## INTRODUCTION

In a world where robotics is becoming more prominent, our ability to trust them has never been so important. With a robot's physical appearance drastically influencing our perceptions of trust, a greater awareness of how design elements and their aesthetic effect may trigger what affective processes are imperative. Robots have an exceptional potential to benefit humans within a team, yet a lack of trust in the robot could result in underutilising or not using the robot at all (*Floyd, Drinkwater & Aha, 2014*). As *Barnes & Jentsch (2010)* identified, the key to a successful relationship between man and machines is in how well they can work and adapt to each other. This can develop through the form and structure of the robot that in turn helps establish social expectations. In addition, a robot's morphology can have an effect on its accessibility and desirability (*Fong, Nourbakhsh & Dautenhahn, 2003*). The research presented in this paper explores how robot aesthetics can heighten participants ability to trust robots. Participants were introduced to an array of robot visualisations (face and chest) and asked to note their impressions towards each visualisation and whether or not they trusted the robot. This enabled the researchers to investigate how design elements and their combined aesthetic arrangement can act as emotional stimuli influencing the ability to trust each robot. In detail, by using various design elements (*i.e.,* colour, blurriness, and tone), we were interested in better understanding how we design for the fundamental principles of aesthetic order in human–robotic interaction. We anticipate that uncertainties in and between the visualisations will greatly influence a participant's willingness to accept the robot (*i.e.,* The cohesion of messages, positive and balanced stimuli, non-invasive colours, *etc.*). This paper highlights not only the impact of risks and uncertainties created by the visualisations on the human–robot interaction but also the potential of robot aesthetics to commence a trusting relationship.

## LITERATURE REVIEW

### Human–robot interaction

Human-Robot Interaction (HRI) is a field dedicated to understanding, designing, and evaluating robotic systems for use by or with humans. (*Huang, 2016*, p.1). *Yanco & Drury (2002)* claim that Human–robot interaction is a subset of the field of human–computer interaction (HCI) and that HRI can be informed by the research in HCI. *Scholtz (2002)* argues that there are many differences between HRI and HCI, dependent on dimensions in the environment, system users and physical awareness. 'The fundamental goal of HRI is to develop the principles and algorithms for robot systems that make them capable of direct, safe, and effective interaction with humans' (*Feli-Siefer & Mataric, 2010*, p.86). It is the 'effective interaction' which is of interest to the authors of this paper (*i.e.,* the ability to build a trusting relationship through effective human–robot interaction). HRI quality may be strongly dependent on the capacity of the communication channel(s) to carry information between human and robot (*Steinfelf et al., 2016*). Robotic communication is

based on three components, the channel of communication, communication cues, and the technology that affects transmission. Information can be communicated through three channels: visual, audio, and environmental (*Billinghurst, Chen & Chase, 2008*). The authors of this paper will be focusing on the visual channel of communication and building affective visual communication cues. A socially interactive robot should be able to communicate its trustworthiness through the use of non-verbal signals including facial expressions and bodily gestures (*Stoeva & Gelautz, 2020*). The face is capable of expressing a range of emotions that others generally have little difficulty identifying (*Landrigan & Silver, 2007*). *Richert et al. (2018)* considers these human-like designs combined with the integration of natural user interfaces could enhance the overall acceptance and interaction of these technologies. In more detail, *Duffy (2003)* states a robot's capacity to be able to engage in meaningful social interaction with people requires a degree of anthropomorphism (human-like qualities). As Gurthrie cited in Daminao & Dumouchel (2018) points out, the tendency to see human faces in ambiguous shapes provides an important advantage to humans, helping them to initially distinguish between friend or enemy and establish an alliance. A robot's appearance at the first interaction can affect how a robot is interpreted by its users, and in turn how the user may interact with the robot (*Luptetti, 2017*). In terms of human–robot interaction the physical appearance can have an important affect (*Canning, Donahue & Scheutz, 2014*), yet before humans are able to effectively interact with robots, they must be able to accept and trust them (*Billings et al., 2012*). This trust is what is of real interest to the authors of this paper, in order to influence how we design for effective trusting relationships between human and robot through their physical and visual appearance.

## Aesthetic interaction

'Aesthetic interaction is not about conveying meaning and direction through uniform models; it is about triggering imagination, it is thought-provoking and encourages people to think differently about interactive systems, what they do and how they might be used differently to serve differentiated goal' (*Petersen et al., 2004*, p.271). Aesthetics can be classified as a core principle of design which encompasses a design's visually pleasing qualities, functionality, and emotional considerations (*Interaction Design Foundation, n.d.*). For many people, an understanding of a robot is achieved through the senses and the reading of bodily form and gestures, facial and chest screens, and sounds as opposed to only the reading of a screen. As a result, it is very important for us to be able to consider the aesthetic processes involved in our interaction with robots. Research shows that aesthetics can afford the construction of associations and meanings through feelings, intuitions, thoughts, memories, *etc.* (whilst we interact with computers), which we can then stitch together to form a deeper understanding and appreciation of what we are seeing/experiencing (*Carroll, 2010*). Indeed, the aesthetic interaction can promote a relationship between the user and the computer (*i.e.,* robot) that encapsulates a person's full relationship—sensory, emotional, and intellectual. In doing so, it can entice an 'engaged interaction' which can change the user's perceptions and interpretations (*Carroll, 2010*). In our human—robotic interactions, the authors of this paper feel that the aesthetic provides many opportunities to enhance our

human—robotic experiences particularly our trust and acceptability of robots. As *Prinz cited in Holmes (2017)* points out, our conscious experience consists of perceptions with shades of feelings—objects (such as robots) can be comforting or scary, sounds are pleasing or annoying, our body feels good or bad—which all can play a crucial role in guiding our behaviours. According to *Moors, Ellsworth & Frijda (2013)*, the basic premise of appraisal theories is that emotions are adaptive responses, which reflect our appraisals of features of the environment/events that are significant for our well-being. Essentially, emotions are elicited by evaluations (appraisals) of how events and situations relate to our important goals, values, and concerns. *Scherer (2009)* suggests that there are four major appraisal objectives that an organism needs to reach to adaptively react to a salient event: relevance (*i.e.,* how relevant is this event for me?), implications (*i.e.,* what are the implications or consequences of this event and how do they affect my well-being, and so on?), coping potential (*i.e.,* how well can I cope with or adjust to these consequences?), and normative significance (*i.e.,* what is the significance of this event for me-concept and for social norms and values?). Interestingly, each emotion has a unique appraisal structure. For example, the aesthetic emotion interest involves two appraisals (*Silvia, 2005*): appraising an event as new, complex, and unfamiliar (a high novelty-complexity appraisal) and as comprehensible (a high coping-potential appraisal). Interest causes an emotional and motivational state that facilitates exploration, engagement, and learning (*Silvia, 2008*); it reflects both the emotional and cognitive aspects of engagement (*Ainley, 2012*). In terms of the aesthetic emotion of knowledge, firstly, the emotions stem from people's appraisals of what they know, what they expect to happen, and what they think they can learn and understand (*Silvia, 2009*). Secondly, the emotions, for the most part, motivate learning, thinking, and exploring, actions that foster the growth of knowledge (*Silvia, 2009*). It is generally agreed that the aesthetic information process starts with input from a stimulus, then continues through several processing stages (*i.e.,* Connected to more profound memorial instances) and ends in the final decision-making (*i.e.,* an evaluative judgement of the stimulus) (*Markovi'c, 2012*). *Locher (2015)* describes the aesthetic experience as occurring in two stages. Firstly, an initial exposure to the artefact where a viewer spontaneously generates a global impression/gist of the work and secondly, where aesthetic processing ensues (*i.e.,* directed focal exploration to expand knowledge and contribute to a viewer's interpretation, aesthetic judgement, and emotions regarding the artefact). *Zajonc (1980)*, claimed that it is possible for us to like something or be afraid of it before we know precisely what it is and perhaps even without knowing what it is. Since this, there have been many researchers who have begun to explore automatic affective processing; the premise is that beings are able to establish good and bad stimulus before establishing contact with the stimulus (*De Houser & Hermans, 2001*). In light of this, the evaluation is subject to the interaction between an event and the appraiser (*Lazarus, 1991*). Importantly, the emotions are elicited according to the way a person appraises a situation (*Ellsworth & Scherer, 2003*). Significantly, however, research shows that certain aesthetic elements can trigger cognitive and affective processes into motion to influence aesthetic appraisals and more especially how a person aesthetically appraises a situation (*Blijlevens, Mugge & Schoormans, 2012*). In fact, stimuli that evoke aesthetic responses are always composites of

multiple elements that do not ordinarily occur together, and when they do, their joint effect is different in kind from the separate effects of the individual elements (*Mechner, 2018*). In terms of visual elements such as colour, line, form, and composition priming certain emotions, *Melcher & Bacci (2013)* found that there is a strong bottom-up and objective aspect to the perception of emotion in abstract artworks that may tap into basic visual mechanisms. In his book, *James (2018)* considered aesthetic emotions to be the immediate and primary sensory pleasure resulting from exposure to a stimulus. Therefore, we ask, can these aesthetic emotions/interactions, in turn, influence how robots are received and how we make decisions to trust them? Indeed, apart from the logical schemes and sense perception, there is also a powerful 'felt' dimension of experience that is prelogical, and that functions importantly in what we think, what we perceive, and how we behave (*Cox &Gendlin, 1963*). What is of real importance to the authors of this paper is the interplay between the aesthetic, cognitive, and affective processes in how we make decisions to trust a robot; in particular, how the in-take of aesthetic information from a robot's facial and/or chest visualisation can influence how we trust the robot.

### Trust, risk and uncertainty

"Trust is a phenomenon that humans use every day to promote interaction and accept risk in situations where only partial information is available, allowing one person to assume that another will behave as expected." (*Cahill et al., 2003*, p.53). For many people, trust is the ability to hold a belief in someone and/or something can be counted upon and dependable, by accepting a level of risk associated with the interaction of another party (*Paradeda et al., 2016*). A willingness to potentially become vulnerable to the actions of others, based on the expectation that the trusted party will perform actions essential or necessary to the trustor (*Mayer, Davis & Schoorman, 1995*). According to *Gambetta (2000)*, trust can be summarised as a particular level of subjective probability with which an agent assesses another in performing a particular action. That trust implicitly means the probability that an action by others will be beneficial enough to consider engaging in cooperation with them despite the risks. Indeed, trust can be evaluated as a probability; however, it is nevertheless a cerebral contract between trustee and trustor that develops within relations between humans (*Coeckelbergh, 2012*). In terms of the robot aesthetic, the authors of this paper feel that we have a unique opportunity to enrich further our knowledge of how designing for trust may afford a unique robotics experience. In situations such as trusting robots where a person's past behaviours and reputations are unknown, we acquire other sources of information to determine a person's motivations (*De Steno et al., 2012*). These other sources of information are used to communicate understanding, which can be done through the use of empathy. As *Lee (2006)* points out, an agent who appears to be empathetic is perceived as more trustworthy, likeable, and caring. Robots do not possess the ability to build traditional relationships with humans; therefore, they rely heavily on visual appearance to portray their trust. As *Lee (2006)* reported, human to human perceptions of trust is widely reliant on the empathy they have for one another. Research shows that a common way in which people convey empathy is in the use of their facial expressions (*Riek & Robinson, 2008*). In robot–human interaction, research

has shown that facial features and expressions can portray important information about others trustworthiness (*Valdesolo, 2013*). For this paper, it highlights the importance of considering the design elements to initiate positive affective processes. Research by *Merritt & Ilgen (2008)*, shows that widespread implementation of automated technologies has required a greater need for automation and human interaction to work harmoniously together. The conclusion has supported that individuals would use machines more if they are trusted than those they do not. It has generally been agreed that where there is trust, there is a risk; as *Gambetta (2000)* indicated, trust is a probability; as you determine the level of risk, you can make alternations to the probability of trustworthiness. *Lewis, Sycara & Walker (2018)* states, the introduction of anthropomorphism poses serious risks, as humans may develop a higher level of trust in a robot than is warranted. Additionally, risks do not always reflect real dangers, but rather culturally framed anxieties originating from social organisation (*Wakeham, 2015*). Interestingly, research by *Robinette et al. (2016)* shows that in certain situations, a person may over-trust a robot while mitigating risks and disregarding the prior performance of the robot. However, another dimension of trust is uncertainty. According to *Wakeham (2015)*, who described being uncertain as having an obscured view of the truth, with a limit on what an individual might know. Uncertainty can cause a restriction in the ability to trust; with uncertainty, you are unable to know all that can happen, resulting in trust becoming a leap of faith (*Nooteboom, 2019*). The decision whether or not to trust a robot based on the uncertainty presented can trigger ethically adjusted behaviours that aim to avoid dangers and minimise potential risk (*Tannert, Elvers & Jandrig, 2007*). Viewing uncertainty from a psychological perspective presents both subjective uncertainty and objective uncertainty. Subjective uncertainty represents a person's feelings, while objective uncertainty is concerned with information a person has (*Schunn & Gregory, 2012*). In more detail, research has shown how uncertainty influences people's ability to trust (*Glaser, 2014*), yet in the same way, trust is a way of dealing with uncertainty and objective risks (*Frederiksen, 2014*).

## MATERIALS AND METHODS

This study was conducted at Cardiff Metropolitan University from the 31st of March 2020 to the 15th of April 2020 and was designed to capture the perception of participants feelings and attitudes towards trusting robots. The study was conducted using the powerful online survey software: Qualtrics. Participants were selected through stratified random sampling to target both participants with past robotic experience and those without. Through distributing the questionnaire on social media, special robotic interest groups, and online forums, the authors were able to obtain participants from a diverse participant pool. A total of seventy-four participants from the age of 16 plus years (50 female & 24 male) completed the study from a varied demographic. Participants resided globally (*i.e.,* Europe, Africa, Asia, Australia, North America, and South America) and captured an assortment of participants. The questionnaire took approximately thirty minutes in duration. All graphics were generated using Adobe Photoshop, and the study and questions asked had a strong aesthetic visual component.

The study mainly consisted of quantitative questions in order to provide summaries through descriptive statistics. Additionally, an assortment of questions required participants to engage in qualitative questions, which then enabled analysis to enrich interpretations and uncover similarities. The questions were separated into two categories to target both the general acceptance of robotics and specific questions relating to the Canbot U03 robot. In order to not influence a participant's feelings and past experience with robots, the Canbot U03 was not shown during the first block of questions. Participants were provided with a brief definition of trust at the start of the questionnaire "To Believe that someone is good and honest and will not harm you, or that something is safe and reliable" (Cambridge Dictionary, 2021, trust entry).

Once participants had concluded the initial preparatory questions, they were introduced to the opening visual of the Canbot U03 robot. Participants were presented to a Canbot U03 (see Fig. 1) with no visual modification and asked whether or not they would trust this robot based on its visual appearance (*i.e.,* only based on the design features). To address the concepts of a participant's ability to trust the Canbot U03 robot, participants were asked to envisage situations in their everyday life where they may encounter a robot. A short list of possible situations and jobs roles were provided to participants (*i.e.,* Teacher, doctor, receptions, bus driver, *etc.*)."

Participants were then presented with a series of questions with different aesthetic modifications throughout. The first modifications came with the Canbot U03 robot presented with a series of cartoon facial expressions portraying different emotions. Participants were prompted to identify the robot's emotion and whether or not they felt the robot was more or less trusting than before. To detail, questions such as the following were asked to participants*: How trustworthy is this robot's appearance?, What emotion do you think the robot is feeling?, Does this visual change affect your ability to trust the robot?, How does the robot make you feel with this appearance?* Participants were also asked to provide descriptions on the following questions: *What characteristics do you believe only robots should have? How do you design a robot that people would trust?*

The following block of questions prompted participants to consider the anthropomorphic characteristics of the robot (see Fig. 2). Participants were introduced to a series of robots that related to having human features; these questions probed participants for their feelings towards these powerful visual modifications.

The next section of questions was related to how the design element colour influenced the participant's opinions and description of the robot. This required participants to associate words (*i.e.,* dangerous, happiest, trusting, unpredictable, and unrealistic) with an array of Canbot U03 robots with different colours hues. Participants were presented with eight robot visualisations (see Fig. 3), all with varying colour hues (*i.e.,* Pink, orange, blue, yellow, *etc.*) and prompted to associate the expressive wording with an individual Canbot, no Canbot or all Canbots.

Participants were also introduced to a range of visualisations with contrasting images such as conflicting facial expressions and chest screen imagery (*i.e.,* Happy facial expression + Danger symbol on the chest). Participants were asked a series of questions such including: *Which Canbot would you describe as most uncertain?, What impact did the*

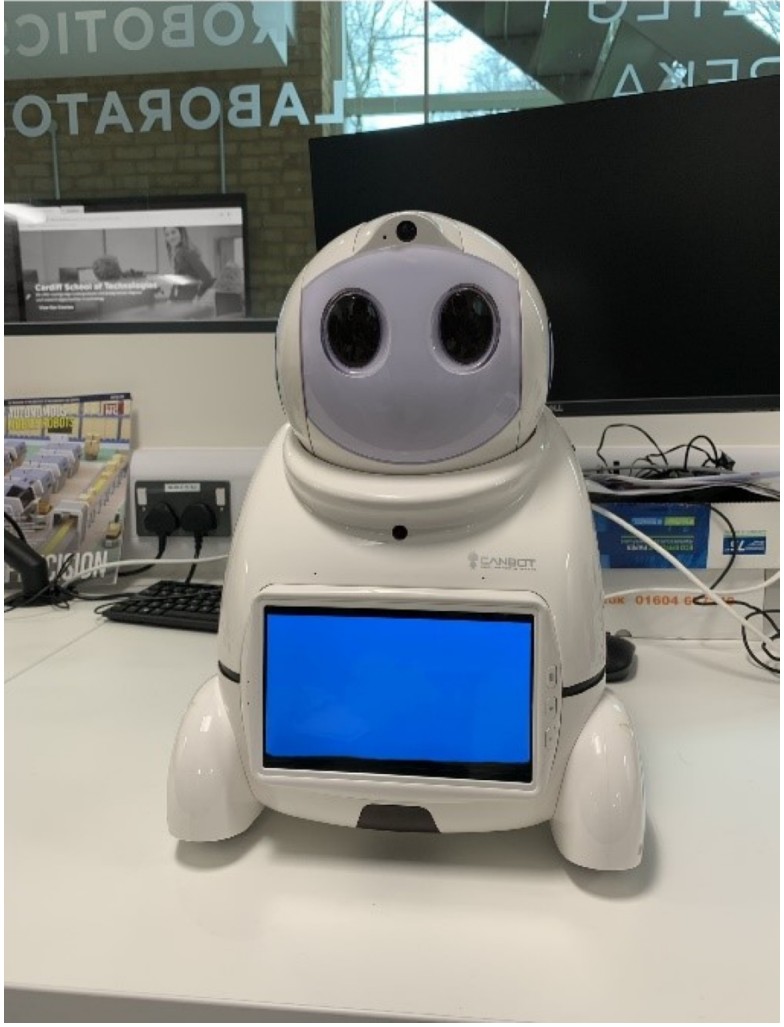

**Figure 1** Original image of the Canbot-U03 robot.

*cohesion of screens have on your decision? and Does the facial expression overrule the icon on the chest screen when considering the Canbot's emotions?* These questions aimed to understand how the level of cohesion between the chest and facial screens can influence a participant's willingness to trust the robot.

Finally, to further probe the concept of risk, participants were presented with mathematical problems that would be too complex for human calculation (*i.e.*, $887 \times 974$ & $997 \times 1{,}066$). Participants were then be asked to identify which Canbot (A–H) displayed the correct solution upon their chest screen. This question required participants to determine the answer they deemed correct based solely on trusting the robot's physical appearance. Optional text boxes were provided throughout the questions to allow participants to expand and express opinions on the robot's appearances.

The Ethics Board at Cardiff Metropolitan University approved the study (CST_2020_Staff_0002), and participants involved were all provided and signed an online

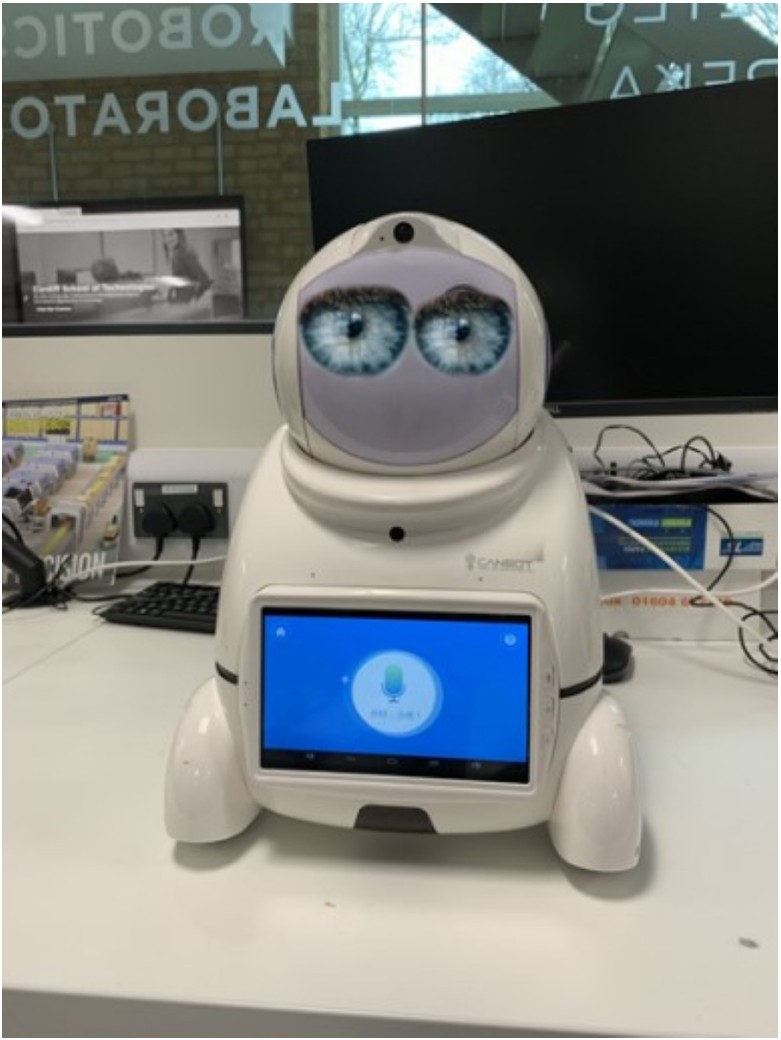

**Figure 2  Canbot-U03 robot with human eyes modification.**

consent form to participate in the study and for the academic use of the non-identifiable data.

## RESULTS

The observations indicate that a participant's willingness to trust a robot was heavily impacted by the aesthetic elements they were exposed to, and whether or not the participant had past experiences with robots. When asked about Fig. 1, fifty per cent of participants said they would trust this robot, twenty-eight per cent were unsure, and the remaining twenty-two per cent recorded that they would not trust the robot. Interestingly, anthropomorphism did not encourage more to trust the robot. Figure 2 (Robot B) shows how the introduction of the face impacted participants who first trusted the robot, twenty of the thirty-seven (fifty-four per cent) of participants who first trusted were now non-trusting or uncertain

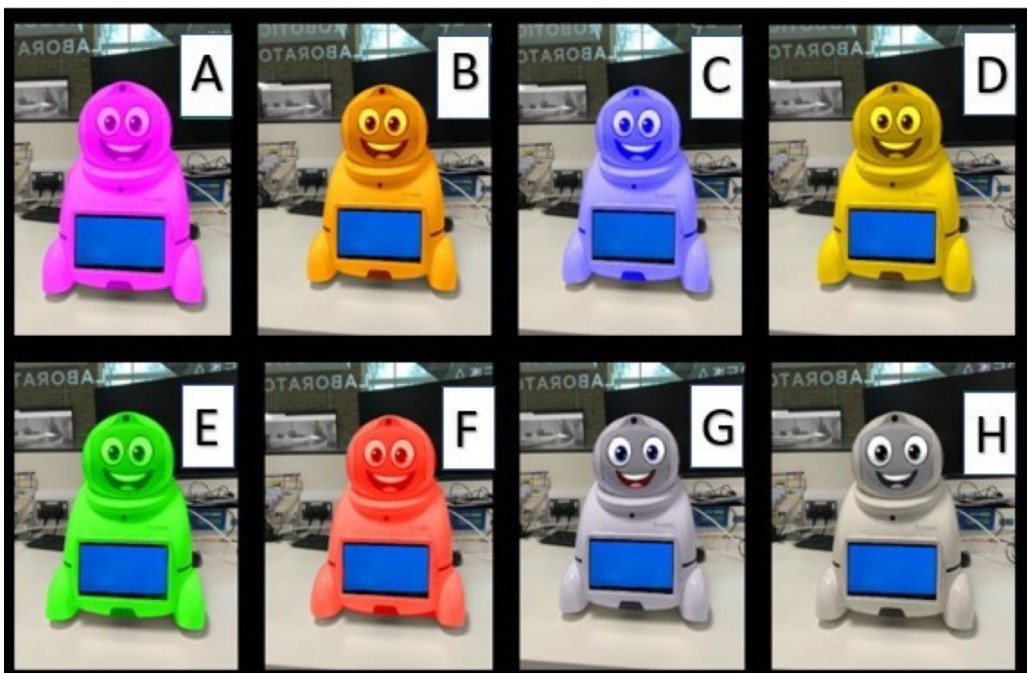

**Figure 3**   Multiple Canbot-U03 robots with different colours hues.

about trusting the robot. However, anthropomorphism did positively influence those unsure of trusting the first robot introduced, with fifty-two per cent changing their opinion from 'unsure' to 'yes' to trust (see Fig. 4). In the human-like visualisations, it seemed participants had different opinions on how robots should be designed for trust. One participant (P72) said, 'Less human-like as this makes them feel more deceptive' while another described human features as 'creepy' and 'People may become intimidated by implementing human behaviours into a machine'.

When probed further into how designing for trust, participants said, 'Give them their own personality that isn't based on human expression 'and that 'human features make the model 'creepy'. One participant notes that the introduction of a realistic human face 'makes people uneasy'. When adding human eyes to the robot visualisation (see Fig. 2) participants were asked their feelings on the realistic eyes. 80 per cent of participants expressed their dislike of this appearance, making them feel 'confused, scared, worried and surprised'. One participant noted 'the need for distinction between human and robot' and 'the inclusion of human likeness may be intimidating'.

When asked **Would you trust this robot?** and **What do you think this robot feels?**

In the blurry face visualisations (Fig. 5), it appeared participants were more apprehensive about trusting the robot. The findings show that half of the participants were able to correctly identify the robot's emotional cue as 'happy' despite the introduction of blurriness. In contrast, the other half of the participants were torn between 'confused, angry, uncertain, uneasy, and uncomfortable' for the robot's emotion. The introduction of the dissimilar stimuli of the happy facial expression and the blurriness presented participants with

| Would you trust this robot? | Robot (A) with no visual changes (original robot) | Robot (B) with enhanced visual changes (Cartoon smiling facial expression) |
|---|---|---|
| YES | 37 | 34 |
| No | 16 | 28 |
| Don't Know | 21 | 12 |

**Figure 4** **Question to participants: would you trust this Canbot with the visual changes?** (A) Indicates participants responses to Canbot with no visual changes. (B) Indicates participants responses to Canbot with smiling cartoon facial expression.

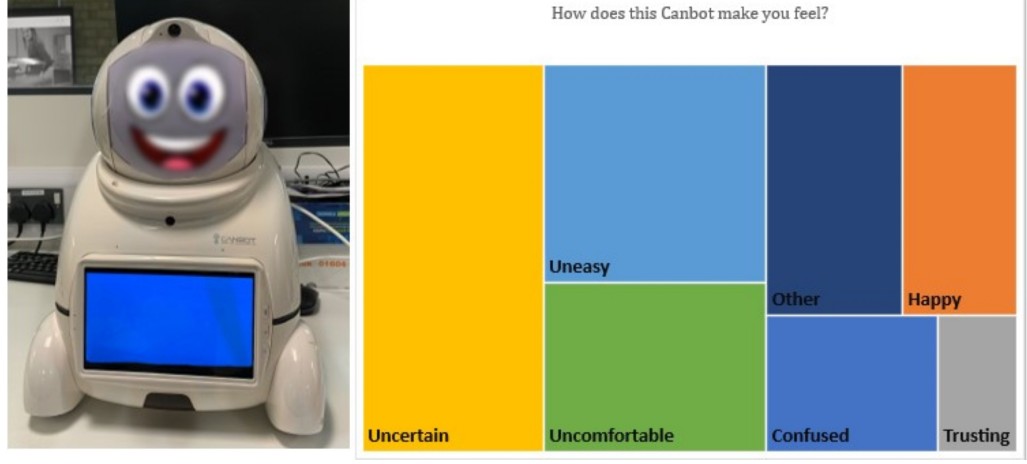

**Figure 5** **Canbot with Blurry facial expression and tree map diagram displaying responses from "How does this Canbot make you feel"?**

uncertainties through the contrasting messages each present (*i.e.,* Happy face – trust, blurriness – uncertain). The results were clearer when prompting participants away from identifying which emotion the robot depicted to how these changes made them feel. The participants concerns were expressed when asked about how the Canbot made them feel, with the majority of responses including terms such as uncertain, uneasy, and confused.

In addition, the findings show different impressions towards facial features when faced with the decision to trust (*i.e.,* What robot is providing you with the correct information?). Interestingly sixty-six per cent of participants selected robot B (Fig. 6) as the most trusting,

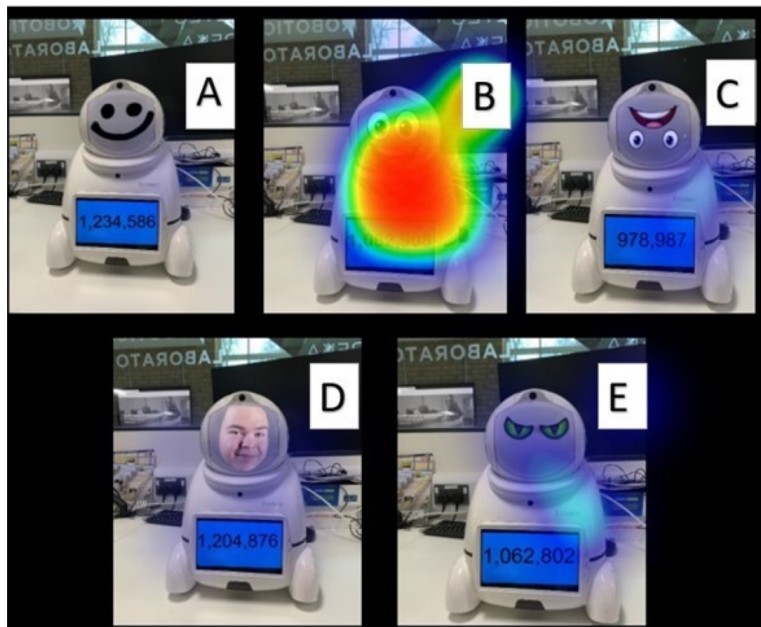
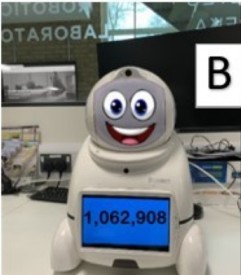

**Figure 6** (A–E) Heatmap displaying participants responses to: what robot would you trust is giving you the correct answer?

despite the introduction of a hybrid robot (Robot D - Fig. 6). Robot five was the next most accepted (fifteen per cent), yet on closer inspection the participant's speed to answer this question was significantly higher (fifty percent increase) than other responses, thus indicating the potential use of a calculator to determine the correct answer to the equation.

Similar results were seen in Fig. 7, with the alterations to the chest screen affording uncertainty to trust the robot to provide the correct answer. We asked participants to indicate which of the six robots posed the correct answer to the 997*1066 by clicking on the chosen robot. Figure 7 displays the frequency distribution of clicks over the six distinct robot images. Sixty-four per cent of participants selected robot B (the robot with limited visual modifications) as most trusting despite providing incorrect information. Interestingly, robot B presented the incorrect answer to the mathematical equation.

Moreover, participants felt that in order for a robot to be trusting, there is a need for 'a screen that clearly shows the message that is being transmitted' and that 'I would expect the screen display to match with any expressions'. In terms of harmony between face and chest screen, one participant highlighted that 'It would be difficult to trust a robot with a face and another image within the robot screen. I would trust better with just one option.' In particular, when exposed to Fig. 8, participants felt that the facial expressions produced a contradicting message to the one upon the chest screen. With sixty per cent of participants declaring the robot as untrustworthy and a further thirty-eight per cent unsure whether or not to trust the robot. One participant could not trust the robot as 'I could not take

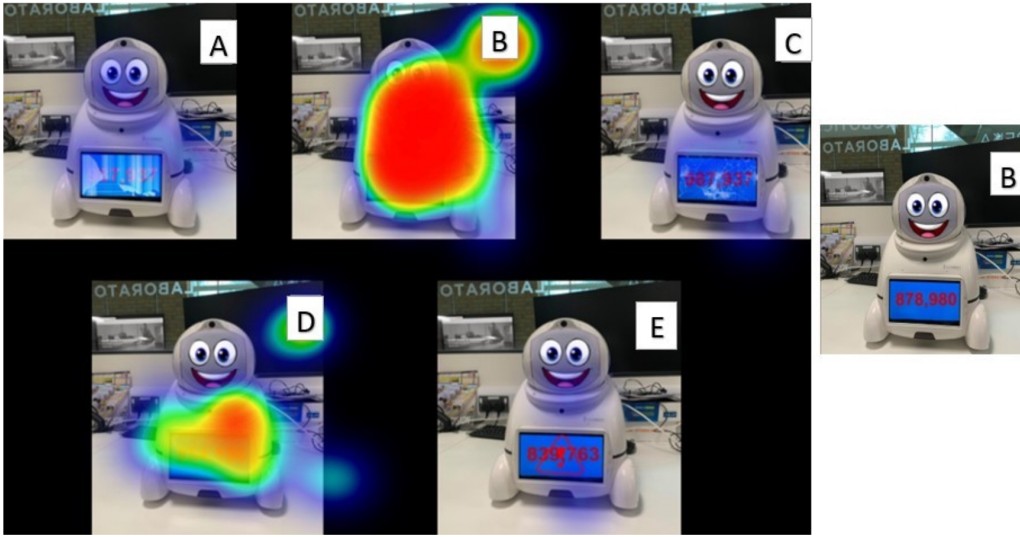

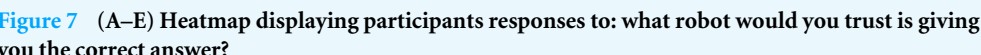

**Figure 7** (A–E) Heatmap displaying participants responses to: what robot would you trust is giving you the correct answer?

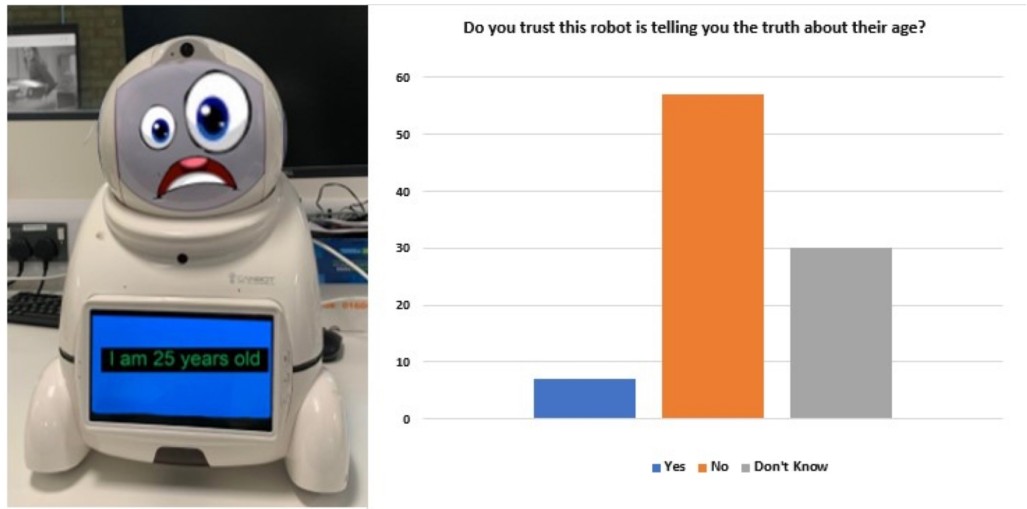

**Figure 8** Robot with confused facial expression and participants responses to: would you trust this robot is telling the truth about their age?

anything this bot says seriously with that expression'. This highlights the true impact of the misaligned messages on participants ability to trust.

## DISCUSSION

In this study, we investigated the impact of the aesthetic order of facial and chest visualisations on participants willingness to trust robots. In particular, it considered the potential risks and uncertainty afforded by certain aesthetic orders to the human–robotic

trusting relationship. Our results show the clear influence that past experience had on a participant's willingness to trust the original robot. Particularly, the visualisation with no modifications was found to have a substantially higher percentage of trust in those with past experience. Participants with no past experiences were relying solely on the visual appearance to determine their level of trust. These findings are in line with what *Sanders et al. (2017)* hypothesised and discovered, in detail, how those participants with past robotics experience would lead to a higher trust of robots and a better positive attitude towards them.

Interestingly, we found that a blurred facial expression significantly influenced whether participants trusted a robot. The blurred facial and chest screen visualisations afforded uncertainty and resulted in a participant's unwillingness to trust a robot.

Figure 7 displayed the extent that physical appearance had on the decision participants made to trust a robot. With a participant's ability to roughly estimate the correct answer not largely adopted by participants, we can only conclude that the physical appearance was the determining factor in the decisions. Interestingly, over half the participants selected robot B, which presented the incorrect answer to the mathematical equation.

Based on previous research that shows colours can influence various moods (*Kurt & Osueke, 2014*), we predicted similarly that the aesthetic element colour could initiate different affective responses when applied to a robot. We tested that hypothesis by introducing participants to an array of robot visualisations that applied an assortment of distinct colour changes. We found that comparably participants were following known psychology of colour associations when selecting what feelings and terms they associated to the robots with the assortment of colours. For example, Fig. 9 displays the words participants associated with the array of colours and other visual modifications. As we hypothesized, certain colours had followed the known associations of related words, such as when participants were promoted to associate the red coloured robot to a particular word. Following the commonly known western culture word associations with the colour red (*i.e.,* dangerous, excitement, festive, *etc.*) (*Cousins, 2012*), we evaluated its affect while present on a robots outer shell and found a similar result of red being associated with the term dangerous.

However, it is important to consider how cultural beliefs and geographical regions may also have an influence on a person's perceptions of colour. A particular colour hue can have multiple meanings and interpretations to people in different regions of the world (*Kurt & Osueke, 2014*). It is critical that when designing a robot to afford trust that these cultural backgrounds, geographical location, and beliefs are carefully considered when selecting a robot's hue to be fit for purpose. Additionally, it is important that this same level of consideration is taken for other design elements, in order to evaluate how the different designs are perceived in different regions, backgrounds, and faiths.

The research has also highlighted the importance of cohesion between the facial screen and chest screen. In the question prompting participants to consider the information on the chest screen (see Fig. 7), the participants were never asked whether or not they trusted the robot as a whole, only if they trusted the information on the screen. However, the negative stimuli released by the facial expression demonstrated that most participants

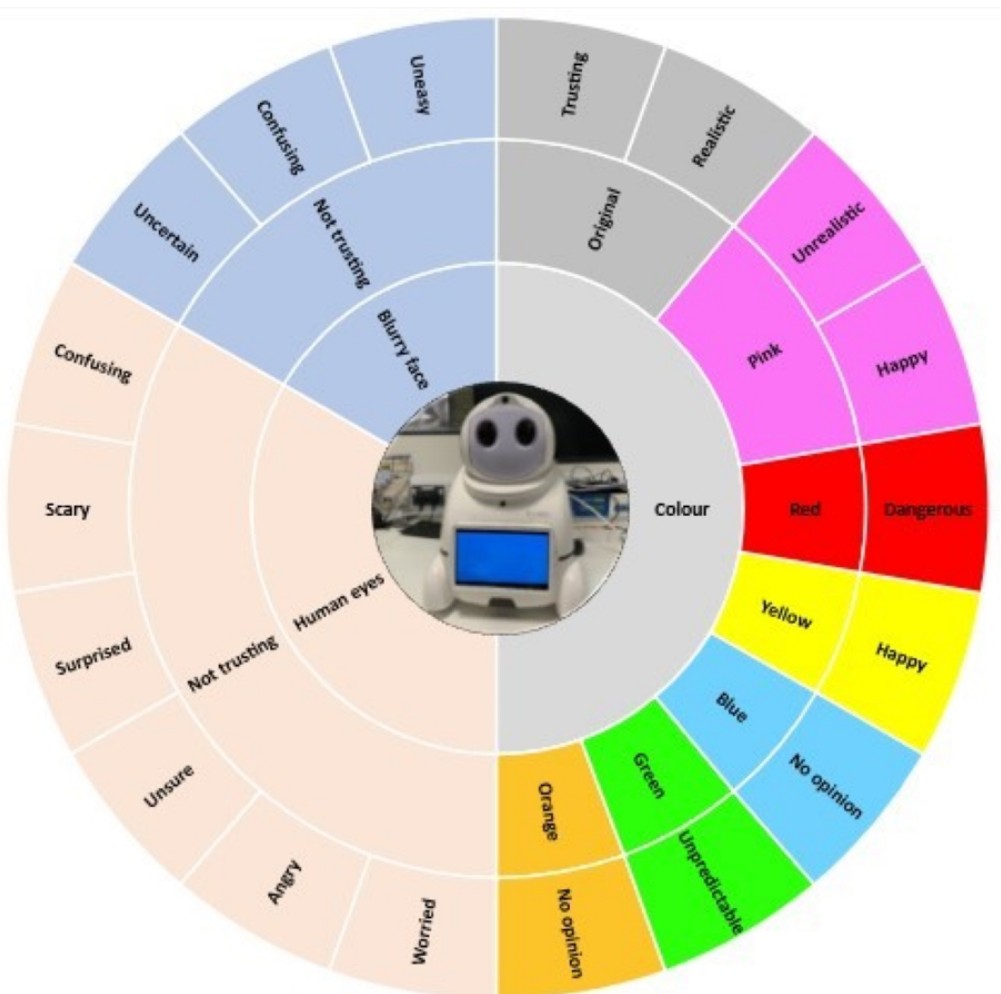

**Figure 9** Sunburst visualisation displaying the visual modifications and participants associated wording.

declared the robot as not trustworthy. Moving forward, when designing a robot that can be trusted, it is important to consider all elements, as stimuli from other visual outputs can potentially influence an independent communication channel.

## CONCLUSION AND FUTURE WORK

This research has shown that robots have the unique ability to create an emotional connection with humans through the use of facial expressions and aesthetics. As documented, we have seen the introduction of anthropomorphism which creates a fine line between increasing trustworthiness and becoming 'scary'. Nevertheless, the non-physical humanlike anthropomorphic designs (cartoon designs) encouraged participants to trust the robots further, showing the unique ability to improve their facilities for empathy. Moreover, this research has shown that the face is not the sole visual aesthetic that can

be utilised to initiate affective processes. The chest screen provides an additional entity to further enrich the potential to provide an engaging experience. Ultimately, the cohesion between the multiple screens is an important consideration for designing socially acceptable robots. As is the design elements and principles to understand how their aesthetic order can play such an important role in initiating a trusting robotic experience.

Going forth, we feel there may be interest in replicating the study but utilising actual robots. We acknowledge there is still a substantial amount of research required to fully understand how we form trusting relationships between human and robot. However, we feel this study paves the way for future studies that involve aesthetic physicalisation, where further sensory cues can be tested to evaluate their influence on our trusting ability of robots. Additionally, this research touched upon how design elements may influence different participants from different cultural backgrounds, geographical locations, and beliefs. We feel it would be of interest to further explore the potential to develop culturally appropriate robots.

Moreover, it would be interesting to further expand on the use of aesthetic designs to evaluate how further modifications (*i.e.,* different colour tones, design elements, design principles, *etc.*) can affect and in some cases, increase a participant's willingness to trust a robot.

Finally, we believe there would be value in understanding how the trusting relationship between human and robot may develop over time. Whilst this study provides details on the initial engagement/interaction, there may be interest to explore aesthetic designs in different situations and time scales.

Throughout this research, we have explored how we can build trusting relationships with robotics through aesthetic designs. In future work, better consideration of human-centered design perspectives must be explored when considering building trust. The research explores participants not trusting robotics as injudicious when the reason not to trust is still a valid and acceptable response in certain situations.

## ACKNOWLEDGEMENTS

We would like to take this opportunity to thank the Eureka Robotics Lab for the loan of the robot used in the study. As one of the flagship research clusters at the Cardiff School of Technologies, EUREKA Robotics Lab is the innovative research hub nested in the School of Technologies, Cardiff Metropolitan University, serving Wales and global stakeholders (*Cardiff Metropolitan University, 2021*).

### Funding

This work is supported by Knowledge Economy Skills Scholarships 2 (KESS2) which is an All Wales higher-level skills initiative led by Bangor University on behalf of the HE sectors in Wales and is funded by the Welsh Government's European Social Fund (ESF)

competitiveness programme for East Wales. There was no additional external funding received for this study. The funders had no role in study design, data collection and analysis, decision to publish, or preparation of the manuscript.

### Grant Disclosures
The following grant information was disclosed by the authors:
The Welsh Government's European Social Fund (ESF).

### Competing Interests
The authors declare there are no competing interests.

### Author Contributions
- Joel Pinney, Fiona Carroll and Paul Newbury conceived and designed the experiments, performed the experiments, analyzed the data, performed the computation work, prepared figures and/or tables, authored or reviewed drafts of the paper, and approved the final draft.

### Ethics
The following information was supplied relating to ethical approvals (i.e., approving body and any reference numbers):
Cardiff Metropolitan University has granted ethical approval to carry out the study (ethics approval number CST_2020_Staff_0002).

### Data Availability
The raw data are available in the Supplemental File.

### Supplemental Information
Supplemental information for this article can be found online at http://dx.doi.org/10.7717/peerj-cs.837#supplemental-information.

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
