# Peer review of "Human-robot interaction: the impact of robotic aesthetics on anticipated human trust"

_PeerJ Computer Science, doi:10.7717/peerj-cs.837_

## Round 0.1 · original submission · Major Revisions

The in-depth reviews suggest a more thorough revision than simply Minor Revisions, despite the brief review to Accept. With that caveat, I am recommending the authors do everything in their power to remedy the two negative reviews, and resubmit for publication reconsideration.

Reviewer 1 ·

Basic reporting

The language of the paper is clear. I spotted a few mistakes that could be removed with an additional edit, but they did not detract much fromt he paper: e.g. line 126 "that that".

The freferences to the literature were appropriate.

The tables and data were all clear.

For me, a major failing of the paper was a failure to examine, even superficially, the concept of "trust", a central concept of the paper. "Do you trust this robot?" but to do what, exactly? In th eonly concrete example given int he paper, it is stated that, "[P]articipants were presented with mathematical 375problems which would be too complex for human calculation (i.e. 887x974 & 997x1066). Participants would then be asked to identify which Canbot (A-H) was displaying the correct solution upon their chest screen. This question required participants to determine the answer they deemed correct based solely on trusting the robot’s physical appearance". The logic of this appears to be that, if the calculation is too complex for a person to complete, a decision could only be based on the appearance of the robot. Even allowing that the calcualtion was "too complex for human calucaltion", a human shoul dbe able to decide whther the corect answer was odd or even, and perhaps have a fair guess at whethe rit was larger or smaller than a million, or even calculate the final digit. So the idea that it could only be the appearance of the robot that determined the outcome seems to me to be deeply flawed.

Experimental design

I have already mentioned some of the shortcomings of the design, in terms of exactly what "trust" was taken to mean.

However, in this section I find the question of sampling and data collection to be of more concern. Respondents to the questionnaire numbered only 74, from all over the world and with a range of different levels of experience of robots. there were obvioulsy too few participants to be able to analyse the differences between groups, but 74 seems much too few to be able to generalise the results to the whole of humanity. So this raises questions about how the sample was selected, what part self-selection palayed, and any resulting biases.

I can see no details in the paper as to how the sample was selected, so it is impossible to be clear about how effectively the process was conducted.

Validity of the findings

In each of the above sections I have commented on weaknesses that I think have a serious impact on the robustness of the findings. I do not feel confident tha tthe evidence provided supports the conlcusions.

·

Basic reporting

This is an interesting and timely paper. It is written in clear, unambiguous language.
Literature references are sufficient to provide background and context for research results. The paper includes sufficient introduction and background to demonstrate how the work fits into the broader field of knowledge. Relevant prior literature should be appropriately referenced. The structure of the article conforms to an acceptable format of ‘standard sections’. The submission is ‘self-contained,’ representing an appropriate ‘unit of publication’, including results relevant to the hypothesis.

Experimental design

The authors provide the reader with original primary research inline with the within aims and scope of the journal. Research question is well defined, relevant & meaningful. It is stated how research fills an identified knowledge gap. The submission clearly defines the research question, which are relevant and meaningful. The knowledge gap investigated is identified, and statements are made explaining how the study contributes to filling that gap.

Validity of the findings

All underlying data have been provided and they are robust and credible. Conclusions are well stated, linked to original research question & limited to supporting results.

Additional comments

This is a timely piece that looks at a novel and interesting research area. Should the authors consider it appropriate, references to complementary disciplines, such as psychology or copyright might be included. Overall this is a good piece of academic writing, ready to be published.

·

Basic reporting

In general, it is a well-written, easy-to read report with clear and sufficient theory and method.
I found a few very small typographical errors (e.g. missing blank space),which I marked in the attached pdf file.
However, I have two larger suggestion for revisions, both concerning the researcher reflection on what they research and in what context.

1. You seem to take for granted that users always should trust robots, and that when users don't trust robots, something is wrong. However, looking at the problem from a human-centered design perspective (or, a critical design perspective as well), we should acknowledge that sometimes technology should not be trusted. So, just an acknowledgement about this, and a reflection on why you take for granted that trust is good, might be good.

2. You have chosen to frame the discussion as a discussion about aesthetics. However, very much of what you have investigated could also be framed as design. Yet, you mention very little about design. Not necessarily an error in the research, but I think it would be good to briefly explain why you have chosen to talk about aesthetics instead of design, and how you see the relation between those two terms (for example, is aesthetics a sub-set of design, or is design a means to reach aesthetics, or even the other way around. While bringing up this, it could also be good to mention that there are very much work done within design when it comes to emotional effects such as trust (trust for brands, trust for other mechanical objects such as cars etc). There is also existing design patterns and design frameworks, for example the gestalt laws, that have been exploring similar questions for many many decades. So an awareness of this might support your results.

Experimental design

In my opinion, a survey is highly limited as a research tool to make statements about trust users would feel towards a robot. The respondents will answer what they THINK they would feel in a real-life situation, or what they THINK you as researchers want them to feel. (Some of the survey questions are quite revealing, it is quite easy for a respondent to second-guess what you as researcher anticipate the respondent to answer.)

If users actually interacted with real-life robots in field studies (real-life situations), then it is possible that their reactions would be quite different from what they state in this survey (as you hint at when you mention further research at the end).

Of course you can't change that now, you have the data you have, and even if I question how much conclusions we can draw, I still think the result you have is worth reporting. But, I strongly suggest that you yourself bring up a discussion about these limitations of your study. Also, when you describe the results, it might be good to choose wording that emphasize that the results are limited, such as "this suggests", "this indicates", andso on.

Validity of the findings

It is difficult to get a clear view of how the survey was designed and how the exact questions where phrased. Some of the Figures seem to show questions, but it is not clear what part of the question is and what is your explanation for me the reader.
I have some issues with the wordings of the survey questions in Figure 4 (If this is the actual survey question, which is as noted not clear). The word “new” is used to describe the right hand image of the robot. Did the respondents see the word “new”. In that case it can have created a bias since some people probably interpret new as better.
Besides this and my comment on 2. Experimental design, I have no further critique or suggestions for revision concerning validity.

Additional comments

I would suggest a change of the title. It is difficult to understand the title before reading the article, and after reading the article, the title doesn't seem to match the actual content of the article (which is probably why it was hard to understand from the start). The article is not about visualization at all, so I strongly suggest taking away that word. Uncertainty is just one of the aspects of trust in robots that are discussed, so I find it strange to include that in the title. I would suggest something like: "Users attitudes concerning trust in robots - towards an understanding of how robot aesthetics impact anticipated trust".

On line 88 you write "between man and machines is in how well they understand each other". It is a philosophical/ontological question maybe, but I object strongly towards the notion that machines "understand" anything. There is no proof as far as I know that any robot or AI actually understands anything, and it will take decades until general intelligence have made this possible. Could this be re-formulated some how (I don't know how it was formulated in Barnes and Jentsch (2010) ) ? Something like "....is how well they adapt to each others behavior", or so. That phrasing doesn't assume any actual understanding.

I don't understand the heatmaps in Figure 6 and 7. Heat maps are used to visualize values distributed in two dimensions. Which value (data) does the heat map represent? Did respondents click on the image, and the heat map shows frequency of clicks? It doesn't make sense, but I can't come up with any other explanation. Please consider if heatmap is correctly used, and either take away or if it is kept explain what it visualize and how.

---

## Round 0.2 · Minor Revisions

Please address reviewer matters for minor revisions and resubmit. Thank you.

Reviewer 1 ·

Basic reporting

I think that the paper could still benefit from another review of the text to remove typographical errors, some of which have been introduced in the course of the modification. For example:
Line 111: There is an unnecessary apostrophe after the full stop.
Line 132: Guthrie should be cited rather than citied.
Line 148: ‘designs’ should have an apostrophe after the ‘n’.
Line 326: Is the question not whether the robot is trustworthy, rather than trusting?
Line 326: ‘robots’ needs an apostrophe after the ‘t’.

Experimental design

The authors have responded to the comments of the reviewers in regard to failures in the descrioption of the experiemntal design, rather than falinings in the desing itself, and this is now much better.

Validity of the findings

In responding to the comments of the reviewers, the authors have now added some clarifications and caveats that strengthen the paper in this regard.

Additional comments

I have to say that I am impressed by the way the authors have responded to the points of criticism raised by all the reviewers, and I would now say that the paper could be published.

I still think there is something lacking in the analysis of trust, and of whether it is the robot that deserves or does not deserve trust. I trust a calculator to produce an answer that accurately reflects the buttons that have been pressed according to its own algorithm. I might doubt my competence in button pressing (fat fingers) or the ease of intuiting the order in which buttons have to be pressed, and therefore have doubts about the answer, but my trust in the calculator would not be diminished. In this study, participants were confronted by robots that had been deliberately programmed to deceive; was it the robots or the programmers who were being assessed as trustworthy? But perhaps this is work for future study.

·

Basic reporting

The amendments introduced in the paper fully reflect my original comments, thank you for incorporating them.

Experimental design

The paper, as amended, meets the PeerJ standards. No specific comment to be added.

Validity of the findings

No further comment.

Additional comments

Thank you for amending the paper, hope the comments have been useful.

·

Basic reporting

The revisions that have been made is a bit shallow and a bit "quick fixes", but they are extensive enough and good enough to make the publication acceptable in my opinion. The authors seems to have understood the feedback, and made appropriate changes.

Experimental design

no comment

Validity of the findings

The limitations and arguments around the findings is more clear now.

Additional comments

No further comments, I will suggest an accept.

Reviewer 4 ·

Basic reporting

Summary: This paper titled “Human-robot interaction: The impact of robotic aesthetics on anticipated human trust” examined people’s preference on some robot faces in terms of facial expression mood, blurriness, and color, as well as conflicting information on the chest display. The authors’ writing is great and clear. However, from a scientific research perspective, the rationale and clarity of the concepts, the experiment design, and the findings all have flaws that make the paper limited contribution to the trust-in-robot literature. Not any of the findings particularly adds new knowledge. Sorry, but I would not recommend publishing this paper.

Background:
1. In the authors’ own words in the introduction, “As Barnes and Jentsch (2010) identified, the key to a successful relationship between man and machines is in how well they can work and adapt to each other,” esthetics of a facial expression is not part of how well they can work and adapt to each other. Trust is a process-based decision-making phenomenon. If the results that the robot gave are wrong, no matter how pretty the face looks, the robot is still untrustworthy.

2. The authors stated, “As Gurthrie citied in Daminao and Dumouchel (2018) points out, the tendency to see human faces in ambiguous shapes provides an important advantage to humans, helping them to distinguish between friend or enemies and establish an alliance.” This is not true. The same person can be a friend at one time and be an enemy at another. It is not the look of the face that changed, but the identity and mechanism changed.

3. The contents of trust have many dimensions, such as trusting the purpose/intention, trusting the process, and trusting the performance. Trust also has different phases, such as initial trust, trained/informed trust, and situational trust. The study is not clean on the direction: Are people willing to trust the robot (Canbot) to do what? If this question is not clear, how does esthetics matter?

Experimental design

Experiment design:
4. The authors stated that “It is the ‘effective’ interaction which is of interest to the authors of this paper (i.e., the ability to build a trusting relationship through effective human-robot interaction).” By definition, interactions mean “mutual or reciprocal action or influence” (Marriam Webster Dictionary). However, the facial expression is just showing the options, and there is no interaction involved. The robots’ answer correctness is not the independent variable for people’s trust in the robot, but the robot’s appearance is the independent variable. The experiment design is not based on interactions.

5. The questions about trust should be more specific. The original question was, “Would you trust Canbot by the way they look?” However, what does it really mean? Trust Canbo on what? Trust Canbo is a robot? Trust Canbo can do all math problems correctly? Trust Canbo provides correct information on all questions? Trust Canbo can take care of your children? Trust Canbo has human conscience? Trust Canbo have the good intention for you?

Validity of the findings

Results and findings:
6. The authors only reported the percentage of participants’ response category, without any statistical significance report at all. More advanced statistics (e.g., non-parametric statistics such as Chi-square in this case) are strongly encouraged.

7. The results that people preferred the good proportioned positive faces (or the visualization with no modifications) better than a weird/angry/blurred/mentally retarded looking is not surprising at all. Is it really trusting in a robot or the one chosen is just least displeasing?

8. The authors stated, “In situations such as trusting robots where a person [should this be a robot?]’s past behaviours and reputations are unknown, we acquire other sources of information to determine a person’s motivations (DeSteno et al., 2012).”, based on this logic, this study is only talking about *the first initial impression of the robots’ trustworthiness*. Once the robot presents one result to a question, that information itself is showing whether the robot is trustworthy. The so-called trust shall not go past the initial impression. The paper should not overgeneralize the findings on the impact of esthetics on trust in a robot.

---

## Round 0.3 · Minor Revisions

There are too many grammar and language inconsistencies and errors in the revision, suggesting the manuscript was not properly proofed. Below are a few such issues:

pick-up -- Not the best word, despite the revision to it.

(cartoon face versus human face etc.) and that certain visual variables (i.e. blur) -- Commas missing here and throughout at times.

In 1980 Zajonc (1980) claimed -- Comma needed, and date is redundant.

(50 Female & 24 Males) 
-- Gender consistency for plurals here, possibly elsewhere.

Human-Robot Interaction (HRI) is a field dedicate to understanding -- Should be dedicated.

(Stoeva and Gelautz, 2020) -- Ampersand expected here and elsewhere.

sunderutilising -- Sp.

colours etc. -- As noted, comma expected.

between friend or enemies -- Gender consistency for plurals here.

(i.e. how relevant is this event for me?), implications (i.e. what are the implications or consequences of this event and how do they affect my well-being, and so on?), coping potential (i.e. how well can I cope with or adjust to these consequences?) and normative significance (i.e. what is the significance of this event for me-concept and for social norms and values?) -- Issue noted above.

(Silvia,2008) -- Too tight here, and elsewhere, space needed.

(Blijlevenset al. -- Space needed.

(Cox and Gendlin, 1963) -- As noted above.

(Paradedaet al., 2016) -- As noted above.

likeable -- Sp.

(i.e., Teacher, doctor, receptions, bus driver etc.) -- Serial comma expected.

As noted, these are just a few examples that render the manuscript in need of final proofing and adjustment before possible acceptance. It does not include other content variables which may be contestable, nor does it include all such issues that are a responsibility of the author(s).

Thank you for your understanding. I look forward to your revision.

---

## Round 0.4 · accepted · Accept

Accepted, but typos and related inconsistencies still need to be addressed before publication.